# Ferric Carboxymaltose in Patients with Acute Decompensated Heart Failure and Iron Deficiency: A Real-Life Study

**DOI:** 10.3390/jpm13081250

**Published:** 2023-08-12

**Authors:** Federico Capone, Alberto Cipriani, Leonardo Molinari, Marianna Noale, Beatrice Gusella, Fabrizio Lucente, Sandro Savino, Antonella Bertomoro, Alois Saller, Sandro Giannini, Roberto Vettor

**Affiliations:** 1Department of Medicine (DIMED), University of Padua, Via Giustiniani, 2, 35128 Padova, Italy; 2Department of Cardiac, Thoracic and Vascular Sciences, Padua University Hospital, 35128 Padua, Italy; 3National Research Council (CNR), Neuroscience Institute, Aging Branch, 35128 Padua, Italy

**Keywords:** acute decompensated heart failure, iron deficiency, ferric carboxymaltose, real-life experience

## Abstract

**Background**: The correction of iron deficiency (ID) with ferric carboxymaltose (FCM) is a recommended intervention in heart failure (HF) with reduced ejection fraction. Our aim is to evaluate, in a real-life setting, the clinical significance of ID screening and FCM treatment in acute decompensated HF (ADHF). **Methods**: In a cohort of ADHF patients, the prevalence of ID and FCM administration were investigated. Among the 104 patients admitted for ADHF, in *n* = 90 (median age 84, 53.5% with preserved left ventricular ejection fraction—LVEF), a complete iron status evaluation was obtained. ID was detected in *n* = 73 (81.1%), 55 of whom were treated with in-hospital FCM. The target dose was reached in *n* = 13. **Results**: No significant differences were detected in terms of age, sex, comorbidities, or LVEF between the FCM-supplemented and -unsupplemented patients. During a median follow-up of 427 days (IQR 405–466) among the FCM-supplemented patients, only 14.5% received FCM after discharge; the mortality and rehospitalizations among FCM-supplemented and -unsupplemented patients were similar (*p* = ns). In a follow-up evaluation, ID was still present in 75.0% of the FCM-supplemented patients and in 69.2% of the unsupplemented patients (*p* = ns). **Conclusions**: In this real-life ADHF cohort, FCM was administered at lower-than-prescribed doses, thus having no impact on ID correction. The significance of our findings is that only achieving the target dose of FCM and pursuing outpatient treatment can correct ID and produce long-term clinical benefits.

## 1. Introduction

Iron deficiency (ID) is common in heart failure (HF) [1,2], and it is associated with an increased hospitalization and mortality risk [3], reduced exercise tolerance, and poor quality of life [4,5], regardless of the presence of anemia. 

The detrimental effects of ID in HF, both in the short- and long-term, include reduced cellular energy generation in the mitochondria and restricted myoglobin production [6,7]. Oral iron supplementation in HF has been shown to be ineffective, possibly due to the coexistence of chronic systemic inflammation, elevated levels of hepcidin, and functional ID [8].

In several clinical trials, intravenous iron supplementation has been shown to be a low-risk intervention capable of improving symptom burden and the quality of life in patients with ID and chronic HF with a reduced ejection fraction (HFrEF) [8,9,10,11,12], yielding the appearance of recommendations regarding ID diagnosis and treatment into international HF guidelines [13,14,15]. A subsequent meta-analysis of four RCTs showed a benefit on recurrent cardiovascular hospitalizations and cardiovascular mortality in patients with ID and HFrEF [16]. 

Despite the consistent body of evidence supporting iron supplementation in these patients, this treatment is seldom administered in every-day clinical practice: the RAID-HF follow-up study demonstrated the impact of ID in chronic HFrEF on long-term (1-year) mortality and quality of life [17], showing that only a minority of patients received iron supplementation in a real-world cohort. Regarding acute decompensated heart failure (ADHF), the AFFIRM-AHF randomized controlled trial demonstrated how supplementation with ferric carboxymaltose (FCM) improves heart failure readmission rates in patients with reduced left ventricle ejection fraction (LVEF) and concomitant ID and ADHF [18]. However, real-life data concerning the impact of iron supplementation on ADHF are lacking. Moreover, clear data on the effects of iron supplementation on ID associated with HF with a preserved ejection fraction (HFpEF) are scant, although there is some evidence of a benefit in this population [19].

### Aim of the Study

The purpose of this study was to perform a real-life analysis of ID screening and iron supplementation in patients with ADHF and ID, with and without anemia. The study further aimed to investigate the effects of in-hospital and follow-up iron supplementation on mortality and readmission rates in patients admitted for ADHF and ID. 

## 2. Methods

This was a retrospective observational study on a real-life cohort of patients hospitalized for ADHF in the Internal Medicine Department of the Azienda Ospedale Università Padova (Italy), between 1 October 2019 and 3 March 2020.

The diagnosis of HF was based on the presence of signs and/or symptoms (declivous edema, pulmonary congestion, jugular swelling, shortness of breath, and reduced exercise tolerance), ultrasound findings (inferior vena cava dilatation and pulmonary congestion), blood tests, and echocardiography. The diagnosis of HFpEF required the presence of structural and/or functional cardiac abnormalities in the presence of a normal (≥50%) LVEF. Cardiac abnormalities included left ventricle (LV) hypertrophy, indirect signs of increased filling pressures (abnormal cardiac chamber enlargement, impaired LV tissue Doppler velocity), or moderate/severe valvular defects. For HFrEF, an LVEF of ≤40% was required, while patients with an LVEF between 41 and 49% were considered affected by HF with mildly reduced EF (HFmrEF). The diagnosis of HF was supported by either elevated circulating levels of natriuretic peptides (NPs) or radiological/hemodynamic evidence of congestion.

Furthermore, we collected clinical data, anamnestic data, and blood tests at admission and hospital discharge.

During the hospital stay, the attending physicians were free to investigate iron deficiency and prescribe iron supplementation. The total dose of FCM administered to each patient was recorded and compared to the expected target dose for that specific patient based on their hemoglobin level and body weight (according to the CONFIRM HF dosing schedule [10]—Appendix A).

The medical history, clinical data, blood test results and imaging findings were collected after informed consent was obtained (as per internal clinical protocol) and managed by the medical staff using electronic data capture tools hosted at the University of Padova. All the clinical investigations were conducted in compliance with the Declaration of Helsinki (2001). The ethical approval was waived by the local Ethics Committee of the University of Padova in view of the retrospective nature of the study, and because all the procedures performed were part of the routine care.

The data collected on admission were: age, sex, body mass index (BMI), Barthel index [20], Charlson Comorbidity Index (CCI) [21], main comorbidities, home therapy, New York Heart Association (NYHA) functional class [22], modified British Medical Research Council (mMRC) questionnaire [23], and LVEF.

The laboratory tests performed during hospitalization were recorded. The first available results among those collected during the hospital stay were included in the analysis. In particular, we assessed the presence of ID (defined as ferritin < 100 μg/L, or 100–299 μg/L with transferrin saturation < 20%) by measuring ferritin and transferrin saturation (TSAT) on admission [15]. According to the WHO criteria, anemia was defined as a hemoglobin (Hb) concentration below 7.5 mmol/L (120 g/L) in non-pregnant women and below 8.1 mmol/L (130 g/L) in men [24].

We subsequently interviewed by telephone patients whose iron status had been assessed during hospitalization. Telephone calls were performed to assess their clinical status and to schedule outpatient evaluations. The collection of clinical information and incident major cardiovascular events using recurrent telephone interviews was preferred over in-person evaluations in consideration of the ongoing COVID-19 pandemic during the follow-up. Between discharge and the follow-up evaluation, patients were treated according to the current clinical practice of the general practitioner/outpatient specialist. The data collected during follow-up were: iron status, iron supplementation after discharge, mortality, readmission for cardiovascular causes, and readmission for all causes. A blood test set was proposed to the responding patients in order to further investigate iron status at the time of follow-up. All the patients found to be iron deficient were offered an additional intravenous supplementation with FCM at our day hospital, according to current indications [25].

### 2.1. Outcomes

The primary endpoint was to evaluate the prevalence of ID in ADHF patients in a real-life survey and to register the frequency and modality of iron supplementation in this setting.

The study also aimed to investigate the effects of in-hospital and follow-up iron supplementation on mortality and readmission rates in patients admitted for ADHF and ID. 

In the latter, we distinguished between readmissions for HF and cardiovascular events (non-fatal stroke, non-fatal myocardial infarction) and all-cause readmissions. 

Further secondary endpoints included: variations in the iron profiles during the follow-up between the supplemented and unsupplemented patients; differences in mortality and readmission rates according to LVEF; differences within and between the groups regarding ferritin, TSAT, and Hb values; and differences in the outcomes between the patients supplemented only during hospitalization and the patients supplemented also after discharge.

### 2.2. Statistical Analysis

Data are presented as the means and standard deviation (SD) or the median (interquartile range (IQR)) for the continuous variables, and as absolute numbers (percentages) for the categorical ones. The normality of the continuous variables distributions was tested through the Shapiro–Wilk test.

The differences in participant characteristics according to ID status, FCM supplementation, and LVEF were evaluated considering Chi-squared or Fisher’s exact tests and generalized linear models, after testing for homoscedasticity (Levene test) or the Wilcoxon rank-sum test, for the categorical and continuous variables, respectively. 

Kaplan–Meier analyses and log-rank tests were performed to evaluate the association between ID and death from all causes, and the association between FCM supplementation in ID patients and hospitalizations (first events, total, and only for HF or cardiovascular events), considering deaths before possible hospitalizations as censored events.

Mixed-effects models were defined to evaluate the changes in ferritin, TSAT, and Hb, according to FCM supplementation and time. Compound symmetry covariance structure and a Tukey adjustment for multiple comparisons were considered. Two-tail *p* values < 0.05 were considered statistically significant. The analyses were performed using the SAS statistical package, version 9.4 (SAS Institute Inc., Cary, NC, USA).

## 3. Results

### Baseline Characteristics

During the study period, 104 ADHF patients were admitted to our medical unit. A complete iron status evaluation (including Hb, ferritin, and TSAT) was obtained for 90 patients (male *n* = 57, 63.3%; mean age 82.7 ± 9.7) that represented our study population (Figure 1). 

A comparison between the baseline characteristics and medications of the patients with or without ID is presented in Table 1. ID was present in 73 patients (81.1%). The prevalence of anemia was 69.9% among the patients with ID and 52.9% in the patients without ID (*p* = 0.18). In 25 patients the LVEF was ≤40% (HFrEF, 29.0%), while in *n* = 15 the LVEF was between 41 and 49% (HF with mildly reduced EF—HFmrEF, 17.5%), and in 46 patients it was ≥50% (HFpEF, 53.5%). There was no statistically significant difference in the demographic (age, sex), clinical (Barthel index, CCI, comorbidities), functional (NYHA, mMRC class) or baseline characteristics when both patient groups were compared (Table 1). No differences were observed for BNP or eGFR between the two groups. Patients suffering from ID had lower Hb levels (113.4 ± 20.1 g/L vs. 127.9 ± 17.0 g/L; *p* = 0.007) than those with normal iron status. 

Among the *n* = 73 patients with ID, 55 (75.3%) were treated with FCM (FCM-supplemented group). The target dose was reached in 13 patients (23.6%) during hospitalization. The clinical characteristics of the FCM-supplemented group versus the FCM-unsupplemented group are presented in Table 2.

Follow-up data were obtained after a median of 427 days (IQR 405–466) from discharge. At the study’s closure (15 May 2021), the fatal and non-fatal outcomes were known for all except for eight patients lost in the follow-up (Figure 1).

Among the 55 patients treated with FCM during their hospital stay, three were lost in the follow-up; eight patients (14.5%) were also supplemented with FCM after discharge. The mean total dose of FCM administered during the study (including outpatient administered doses) was 854 mg. When comparing the FCM-supplemented with the FCM-unsupplemented group, no statistical significance in mortality was found: in the FCM-supplemented group, a total of *n* = 10 (19.2%) deaths occurred; *n* = 3 (20.1%) deaths were observed in the ID-unsupplemented group (Table 3).

A survival analysis related to death for all causes showed no difference between the two groups (patients with ID vs. patients without ID, Log Rank *p* = ns). Readmission for HF or cardiovascular events occurred in 26 out of 52 (50.0%) ID-patients supplemented at the time of recruitment, and in 5 out of 15 (33.3%) ID-patients unsupplemented (Log Rank *p* = ns). No statistical differences were observed for both “all cause” (Figure 2) and “HF or cardiovascular events” readmissions (Figure 3) between the supplemented and unsupplemented patients. 

Patients with ID who were treated with FCM during their hospital stay showed a longer hospitalization compared to the unsupplemented group (median of 6 days vs. 11 days, *p* = 0.0002). A sub-analysis did not reveal any statistical differences in the outcomes between patients with LVEF ≤ 50% and the HFpEF group, regardless of whether the patients were supplemented with FCM during hospitalization and/or follow-up (Appendix A).

During follow-up, a blood test set was obtained from *n* = 35 patients (38.9%), of which *n* = 24 (26.6%) belonged to the FCM-supplemented group. The median time between iron supplementation and blood test evaluation was 438 days (IQR 401–465). The estimated mean change from baseline to week 61 in Hb was 13.8 g/L in the FCM-supplemented group and 2.7 g/L in the unsupplemented group (Table 4). 

Compared to the unsupplemented group, both serum ferritin (Appendix A) and TSAT (Appendix A) increased in the supplemented group. Ferritin increased by a mean of 78 µg/L in the patients supplemented during follow-up (*p* = 0.0015); the median ferritin decreased by 124 µg/L in the patients that did not receive iron supplementation during follow-up (*p* < 0.0001). The TSAT increased by 8.7% in the FCM-supplemented group and by 3.7% in the unsupplemented group during follow-up (*p* = 0.026); no significant differences were obtained in the unsupplemented group. Nevertheless, ID was present in 75.0% of the FCM-supplemented patients and 69.2% of the unsupplemented patients (*p* = ns).

Examining in-depth the effect of follow-up supplementation, independently from the in-hospital administration of FCM, we found that among the ID patients (*n* = 67), only eight were supplemented during follow-up. Among the 59 patients who did not receive FCM during follow-up, 12 died. On the contrary, all the outpatients to whom iron was administered survived (*p* = 0.013). 

## 4. Discussion

ID is associated with worse outcomes in patients suffering from HF. Iron supplementation has been proven to be effective in improving symptoms, mortality, and readmission rates in patients with HFrEF, both in acute and chronic settings. The findings, nevertheless, mainly come from randomized controlled trials. Real-life data, especially regarding the effects on patients hospitalized for ADHF, are scant. We aimed to investigate how often ID screening and FCM administration were conducted in patients admitted for ADHF in clinical practice. Secondarily, we analyzed how effective the observed FCM administration was in reducing readmission rates and mortality in clinical practice. 

We conducted a real-life analysis, tracing a cohort of 104 patients admitted for ADHF, 90 of whom received a complete evaluation of their iron status. Half of them had HFpEF. Among the vast majority of patients affected by ID (81.1% of the study population), three-quarters (*n* = 55) were treated with iron (FCM-supplemented group); the remaining were not (FCM-unsupplemented group). However, the target dose was only administered in *n* = 13 (23.6%). After a year of follow-up, our main findings are: the FCM-supplemented patients received an 854 mg mean dose of FCM and, after a median follow-up of 427 days, this resulted in significantly higher levels of ferritin, TSAT, and Hb. This improvement in iron storage, however, was not enough to significantly exceed the threshold of ID: the prevalence of ID was similar in the FCM-supplemented and -unsupplemented patients at the end of the follow-up. These results are significant, as they reveal that in real life, patients are often treated with FCM only during hospitalization, so no significant improvement in long-term iron status is achieved.

The FCM-supplemented and FCM-unsupplemented groups showed similar mortality and readmissions for HF or cardiovascular events. A subgroup analysis showed no difference in the outcome between patients with a LVEF above or below 50%. The FCM-supplemented patients showed a longer length of stay compared to the unsupplemented ones. This was maybe due to a more severe burden of comorbidities in the supplemented patients (a higher proportion suffered from COPD and their Hb levels were lower, Table 2), even if the HF severity was similar.

Why was iron supplementation not effective in improving the outcomes in our cohort? Several explanations may be provided. First of all, the prescribed target dose of FCM [25] was only reached in a minority of our patients. Target-dose achievement often requires a double injection of FCM with a 7-day minimum timespan between the two administrations. The median length of stay in our cohort was 9 days; this prevented a full-dose correction during hospitalization in several cases. Moreover, ID was often neglected during the long-term follow-up: only 14.5% of the in-hospital FCM-supplemented patients received iron after discharge. A lack of awareness of the importance of ID in HF by general practitioners and the absence of a dedicated service for outpatient iron administration limited the prescription of FCM injections after discharge. This resulted in a non-significant reduction in ID prevalence at the end of the follow-up in the FCM-supplemented patients. The persistence of ID in the long-term most likely contributed to the ineffectiveness of FCM supplementation in determining an outcome improvement. ID is undoubtedly a chronic condition in patients suffering from an inflammatory condition, as HF is proven to be. An isolated, underpowered iron administration can only result in the persistence of ID: if ID persists, an outcome improvement cannot be achieved. 

When comparing our population with the larger and more heterogeneous one recruited in the AFFIRM-HF trial, the first main difference to highlight concerns sample size. Our study was designed to provide a snapshot of the current clinical practice in ID screening and supplementation in Western country referral hospitals. The sample size was not sufficient to provide enough statistical power for a survival analysis and/or recurrent hospitalization comparison. Nevertheless, the data presented here highlight the importance of proper ID supplementation to achieve a reasonable and cost-effective Number Needed to Treat (NNT). The second difference with the AFFIRM-HF trial is LVEF. The patients enrolled in the trial were all affected by HFrEF. On the contrary, only half of the patients included in our analysis showed an LVEF < 50%. Despite not being endorsed by any of the most recent HF guidelines [15], in our institution it was decided to also administer FCM in HFpEF patients. The data from the literature are not unequivocal on the benefits resulting from iron supplementation in these patients [19]. Therefore, the lack of effect may be attributed to the presence of patients with a preserved LVEF in our study population. 

Some limitations of our work must be acknowledged. The monocentric nature of the observation and the small sample size limit the generalizability of our findings. A significant number of variables have been evaluated to compare the baseline characteristics of the FCM-supplemented and -unsupplemented patients. Except for a statistically significant (but probably, not clinically relevant) higher proportion of COPD patients in the first group, the two populations are comparable. However, it cannot be excluded that unconsidered confounding factors played a role in deflecting the onset of outcome, both during hospitalization and during follow-up.

## 5. Conclusions

In our real-life cohort, despite an effective ID screening, intravenous iron supplementation was correctly performed only in a minority of iron-deficient ADHF patients. This resulted in a non-significant improvement in ID status after a one-year-long follow-up. Our study was not designed to fully investigate the effect of FCM on mortality or readmission rates, so we cannot make solid conclusions about the effect of the drug on these outcomes. However, it should be noted that in our cohort, FCM administration in ADHF patients with both HFrEF and HFpEF did not improve mortality or readmission rates. This observation may be due to the small sample size, suggesting a high number needing treatment, or an insufficient total iron dose, emphasizing how important it is to achieve the target dose. Furthermore, the ineffectiveness of FCM administration in patients with HFpEF may merit further investigation in properly designed clinical trials. Our findings show that administering an in-hospital low dose of FCM may only result in higher costs without a clinical benefit. Reaching the target dose appears to be essential to improve outcomes. An outpatient dedicated service for identification screening and supplementation in HF is probably crucial to achieve a long-term ID correction, a needed stepstone for mortality and readmission-rate improvement. 

## Figures and Tables

**Figure 1 jpm-13-01250-f001:**
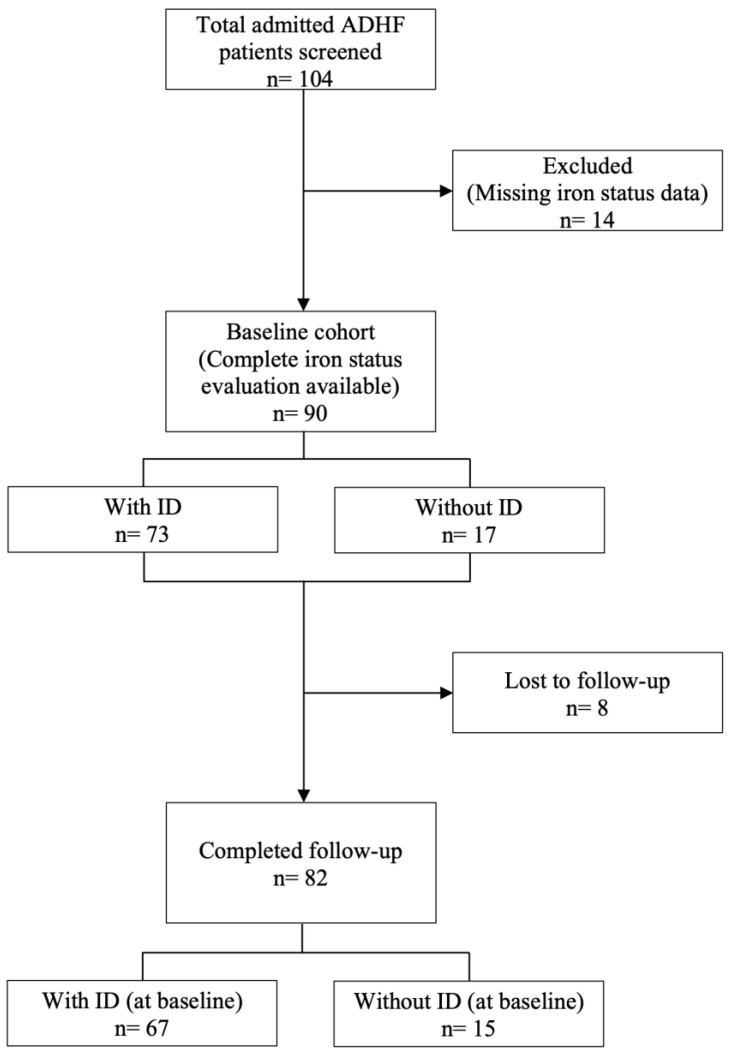
STROBE diagram of the study population.

**Figure 2 jpm-13-01250-f002:**
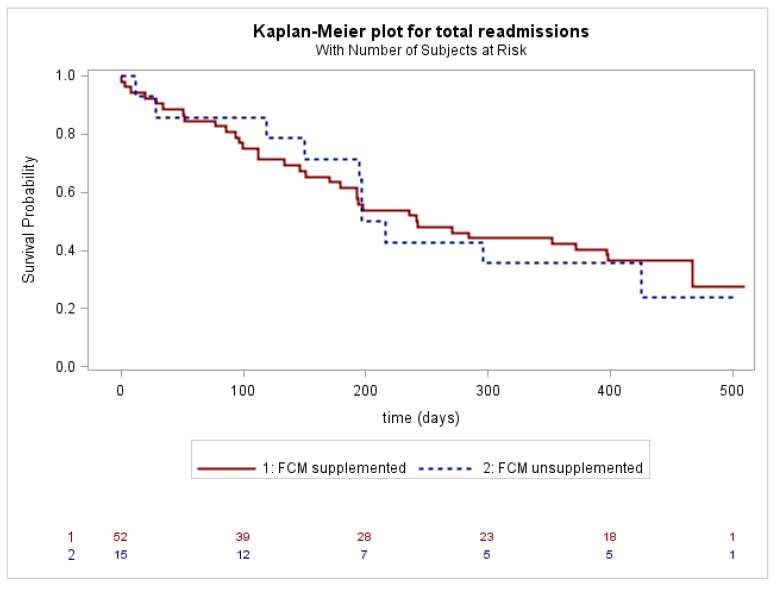
Total readmissions in ID-patients supplemented with FCM (solid line) or unsupplemented (dashed line).

**Figure 3 jpm-13-01250-f003:**
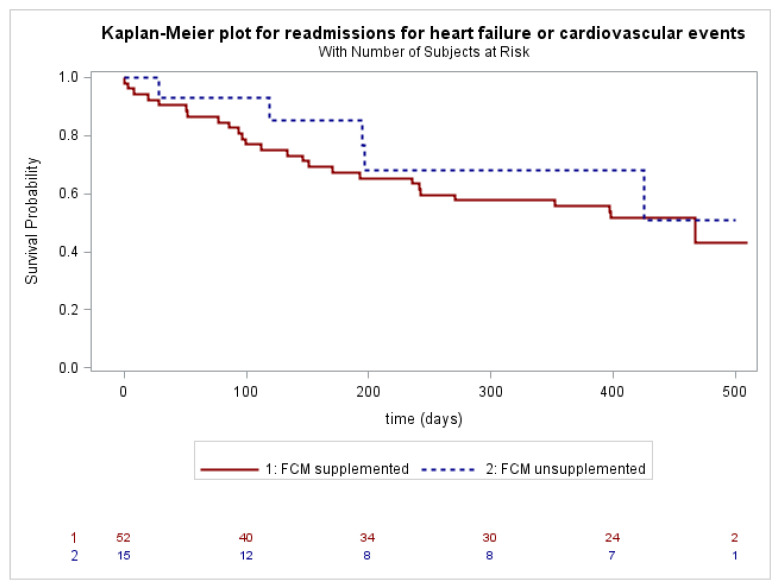
Readmissions for heart failure or cardiovascular events in ID-patients supplemented with FCM (solid line) or unsupplemented (dashed line).

**Table 1 jpm-13-01250-t001:** Baseline characteristics on admission, comparing patients with and without ID.

	(*n* = 90)	With ID (*n* = 73)	Without ID (*n* = 17)	*p* Value
Sex, *n* (%)				0.2120
Male	57 (63.3)	44 (60.3)	13 (76.5)	
Female	33 (36.7)	29 (39.7)	4 (23.5)	
Age (years), mean ± SD	82.7 ± 9.7	82.9 ± 8.9	82.1 ± 12.7	0.7722
BMI, kg/m^2^, mean ± SD	27.1 ± 5.9	26.9 ± 5.7	28.2 ± 6.9	0.4822
Barthel index, mean ± SD	48.5 ± 25.5	49.1 ± 27.0	45.9 ± 18.6	0.6447
CCI, *n* (%)				0.6928
0	0 (0.0)	0 (0.0)	0 (0.0)	
1	12 (13.5)	9 (12.5)	3 (17.7)	
2+	77 (86.5)	63 (87.5)	14 (82.4)	
Comorbidities, *n* (%)				
Coronary artery disease	46 (51.1)	39 (53.4)	7 (41.2)	0.3629
Hypertension	72 (80.0)	57 (78.1)	15 (88.2)	0.5068
COPD	45 (50.0)	35 (48.0)	10 (58.8)	0.4191
Diabetes Mellitus	35 (38.9)	28 (38.4)	7 (41.2)	0.8299
Malignancy	24 (26.7)	20 (27.4)	4 (23.5)	1.0000
AF/AFL	21 (23.3)	17 (23.3)	4 (23.5)	1.0000
Chronic Kidney Disease	16 (17.8)	12 (16.4)	4 (23.5)	0.4925
Medications, *n* (%)				
ACEi/ARB	41 (45.6)	34 (46.6)	7 (41.2)	0.6873
Betablockers	55 (61.1)	46 (63.0)	9 (52.9)	0.4429
Loop diuretics	63 (70.0)	52 (71.2)	11 (64.7)	0.5969
MRAs	27 (30.0)	21 (28.8)	6 (35.3)	0.5969
ARNi	2 (2.2)	0 (0.0)	2 (11.8)	**0.0340**
Digoxin	3 (3.3)	3 (4.7)	0 (0.0)	1.0000
Antiplatelets	35 (38.9)	30 (41.1)	5 (29.4)	0.3735
Anticoagulants	52 (57.8)	41 (56.2)	11 (64.7)	0.5215
NYHA, *n* (%)				0.2427
1	3 (3.5)	2 (2.8)	1 (6.3)	
2	15 (17.2)	12 (16.9)	3 (18.8)	
3	34 (39.1)	31 (43.7)	3 (18.8)	
4	35 (40.2)	26 (36.6)	9 (56.3)	
mMRC, *n* (%)				0.7758
1	1 (1.2)	1 (1.5)	0 (0.0)	
2	3 (3.7)	3 (4.5)	0 (0.0)	
3	6 (7.4)	6 (9.0)	0 (0.0)	
4	16 (19.8)	13 (19.4)	3 (21.4)	
5	38 (46.9)	29 (43.3)	9 (64.3)	
6	17 (21.0)	15 (22.4)	2 (14.3)	
LVEF, *n* (%)				0.5573
>50%	46 (53.5)	39 (55.7)	7 (43.8)	
40–50%	15 (17.5)	13 (18.6)	2 (12.5)	
30–40%	15 (17.4)	11 (15.7)	4 (25.0)	
<30%	10 (11.6)	7 (10.0)	3 (18.8)	
LVEF, *n* (%)				0.3867
>50%	46 (53.5)	39 (55.7)	7 (43.8)	
≤50%	40 (46.5)	31 (44.3)	9 (56.3)	
Hb (g/L), mean ±SD	116.1 ± 20.3	113.4 ± 20.1	127.9 ± 17.0	**0.0071**
Hb < 120 (F) o <130 (M) g/L, *n* (%)	60 (66.7)	51 (69.9)	9 (52.9)	0.1825
BNP, ng/L, median (IQR)	786 (458, 1500)	801 (484, 1511)	545 (275, 1463)	0.1199
eGFR, ml/min/1.73 m^2^, median (IQR)	50.5 (37.0, 65.4)	49.8 (36.3, 64.0)	64.0 (46.8, 81.6)	0.1504
HbA1c, mmol/mol, median (IQR)	44.0 (39.0, 52.0)	45.0 (39.0, 52.0)	42.5 (37.0, 48.0)	0.2913
TSH, mIU/L, median (IQR)	1.53 (0.91, 2.33)	1.44 (0.88, 2.38)	1.69 (1.20, 2.09)	0.7245
Ferritin, ug/L, median (IQR)	70.0 (34.0, 225.0)	58.0 (27.0, 116)	469.0 (299.0, 684)	**<0.0001**
Serum iron, umol/L, median (IQR)	6.1 (4.3, 9.2)	6.0 (4.2, 8.0)	10.7 (7.1, 11.2)	**0.0037**
TSAT, %, median (IQR)	10.3 (7.1, 16.3)	9.6 (6.0, 12.5)	18.5 (12.3, 22.8)	**0.0005**

ID: iron deficiency; SD: standard deviation; BMI: body mass index; CCI: Charlson Comorbidity Index; COPD: chronic obstructive pulmonary disease; AF: atrial fibrillation; AFL: atrial flutter; ACEi: angiotensin-converting enzyme inhibitors; ARB: angiotensin receptor blockers; MRAs: mineralocorticoid receptor antagonists; ARNi: angiotensin receptor–neprilysin inhibitors; NYHA: New York Heart Association; mMRC: modified British Medical Research Council (mMRC) (questionnaire); LVEF: left ventricular ejection fraction; Hb: hemoglobin; BNP: brain natriuretic peptide; eGFR: estimated glomerular filtration rate; HbA1c: hemoglobin A1c; TSH: thyroid-stimulating hormone; TSAT: transferrin saturation. Bold: highlight significant *p* values.

**Table 2 jpm-13-01250-t002:** Baseline characteristics on admission, comparing the FCM-supplemented and -unsupplemented ID patients.

	(*n* = 73)	FCM Supplemented (*n* = 55)	FCM Unsupplemented (*n* = 18)	*p* Value
Sex, *n* (%)				0.2327
Male	44 (60.3)	31 (56.4)	13 (72.2)	
Female	29 (39.7)	24 (43.6)	5 (27.8)	
Age (years), mean ± SD	82.9 ± 8.9	83.0 ± 8.4	82.6 ± 10.7	0.8618
BMI, kg/m^2^, mean ± SD	26.9 ± 5.7	27.3 ± 6.0	25.9 ± 4.9	0.3873
Barthel index, mean ± SD	49.1 ± 27.0	45.4 ± 25.6	60.9 ± 28.6	**0.0377**
CCI, *n* (%)				1.0000
0	0 (0.0)	0 (0.0)	0 (0.0)	
1	9 (12.5)	7 (12.7)	2 (11.8)	
2+	63 (87.5)	48 (87.3)	15 (88.2)	
Comorbidities, *n* (%)				
Coronary artery disease	39 (53.4)	28 (50.9)	11 (61.1)	0.4513
Hypertension	57 (78.1)	43 (78.2)	14 (77.8)	1.0000
COPD	35 (48.0)	30 (34.6)	5 (27.8)	**0.0485**
Diabetes Mellitus	28 (38.4)	23 (41.8)	5 (27.8)	0.2876
Malignancy	20 (27.4)	15 (27.3)	5 (27.8)	1.0000
AF/AFL	17 (23.3)	14 (25.5)	3 (16.7)	0.5361
Chronic Kidney Disease	12 (16.4)	8 (14.6)	4 (22.2)	0.4744
Medications, *n* (%)				
ACEi/ARB	34 (46.6)	26 (47.3)	8 (44.4)	0.8346
Betablockers	46 (63.0)	37 (67.3)	9 (50.0)	0.1876
Loop diuretics	52 (71.2)	42 (76.4)	10 (55.6)	0.0905
MRAs	21 (28.8)	16 (29.1)	5 (27.8)	0.9149
ARNi	0 (0.0)	0 (0.0)	0 (0.0)	--
Digoxin	3 (4.7)	3 (6.0)	0 (0.0)	1.0000
NYHA, *n* (%)				0.7788
1	2 (2.8)	1 (1.9)	1 (5.9)	
2	12 (16.9)	9 (16.7)	3 (17.7)	
3	31 (43.7)	24 (44.4)	7 (41.2)	
4	26 (36.6)	20 (37.0)	6 (35.3)	
mMRC, *n* (%)				0.6897
1	1 (1.5)	1 (2.0)	0 (0.0)	
2	3 (4.5)	2 (3.9)	1 (6.3)	
3	6 (9.0)	5 (9.8)	1 (6.3)	
4	13 (19.4)	8 (15.7)	5 (31.3)	
5	29 (43.3)	24 (47.1)	5 (31.3)	
6	15 (22.4)	11 (21.6)	4 (25.0)	
EF, *n* (%)				0.1234
>50%	39 (55.7)	31 (58.5)	8 (47.1)	
40–50%	13 (18.6)	9 (17.0)	4 (23.5)	
30–40%	11 (15.7)	10 (18.9)	1 (5.9)	
<30%	7 (10.0)	3 (5.7)	4 (23.5)	
EF, *n* (%)				0.4090
>50%	39 (55.7)	31 (58.5)	8 (47.1)	
≤50%	31 (44.3)	22 (41.5)	9 (52.9)	
Hb (g/L), mean ± SD	113.4 ± 20.1	110.1 ± 20.7	123.4 ± 14.5	**0.0136**
Hb < 120 (F) o < 130 (M) g/L, *n* (%)	51 (69.9)	41 (74.6)	10 (55.6)	0.1275
BNP, ng/L, median (IQR)	801 (484, 1511)	798 (463, 1442)	878 (526, 3775)	0.2207
eGFR, ml/min/1.73 m^2^, median (IQR)	49.8 (36.3, 64.0)	49.3 (35.9, 64.8)	50.3 (41.2, 63.2)	0.7460
HbA1c, %, median (IQR)	45.0 (39.0, 52.0)	44.0 (39.0, 53.0)	46.0 (42.0, 49.0)	0.8174
TSH, mIU/L, median (IQR)	1.44 (0.88, 2.38)	1.35 (0.86, 2.19)	1.89 (0.97, 3.00)	0.2871
Ferritin, µg/L, median (IQR)	58.0 (27.0, 116)	51.5 (27.0, 104)	93.0 (39.0, 143)	0.2937
Serum iron, umol/L, median (IQR)	6.0 (4.2, 8.0)	5.8 (4.0, 6.8)	8.1 (5.9, 10.3)	**0.0308**
TSAT, %, median (IQR)	9.6 (6.0, 12.5)	8.7 (5.0, 11.6)	12.8 (10.6, 19.6)	**0.0019**

ID: iron deficiency; SD: standard deviation; BMI: body mass index; CCI: Charlson Comorbidity Index; COPD: chronic obstructive pulmonary disease; AF: atrial fibrillation; AFL: atrial flutter; ACEi: angiotensin-converting enzyme inhibitors; ARB: angiotensin receptor blockers; MRAs: mineralocorticoid receptor antagonists; ARNi: angiotensin receptor–neprilysin inhibitors; NYHA: New York Heart Association; mMRC: modified British Medical Research Council (mMRC) (questionnaire); EF: ejection fraction; Hb: hemoglobin; BNP: brain natriuretic peptide; eGFR: estimated glomerular filtration rate; HbA1c: hemoglobin A1c; TSH: thyroid-stimulating hormone; TSAT: transferrin saturation. Bold: highlight significant *p* values.

**Table 3 jpm-13-01250-t003:** Follow-up results of patients with and without ID; comparison between the FCM-supplemented and FCM-unsupplemented subgroups among patients with ID.

	With ID (*n* = 67)	Without ID (*n* = 15)	*p* Value	FCM Supplemented (*n* = 52)	FCM Unsupplemented (*n* = 15)	*p* Value
Death, *n* (%)	13 (19.4)	2 (13.3)	0.7228	10 (19.2)	3 (20.0)	1.0000
All-cause readmissions, *n* (%)	44 (65.7)	11 (73.3)	0.8150	34 (65.4)	10 (66.7)	0.9266
Readmissions for heart failure or cardiovascular events, *n* (%)	31 (46.3)	8 (53.3)	0.7882	26 (50.0)	5 (33.3)	0.2541
LOS (days), median (Q1, Q3)	9 (7, 13)	9 (7, 13)	0.4017	11 (8, 15)	6 (5, 9)	**0.0002**

ID: iron deficiency; FCM: ferric carboxymaltose; LOS: length of stay. Bold: highlight significant *p* values.

**Table 4 jpm-13-01250-t004:** Estimated within-group and between-group differences in ferritin, TSAT, and Hb among ID patients; comparison between the FCM-supplemented vs. FCM-unsupplemented groups.

	Mean Differences within Group								Mean Differences between Groups (Follow-Up)		
	FCM Supplemented				FCM Unsupplemented						
	Admission (SE)	Follow-Up (SE)	Mean Difference (SE)	*p* Value	Admission (SE)	Follow-Up (SE)	Mean Difference (SE)	*p* Value	Mean Difference (SE)	*p* Value ^§^	Effect Size d
Ferritin, ug/L	143.8 (10.7)	222.7 (17.4)	78.9 (20.8)	**0.0015**	180.7 (14.6)	56.7 (22.4)	−124.0 (27.1)	**<0.0001**	−166.1 (28.9)	**<0.0001**	0.63
TSAT, %	10.7 (0.7)	19.4 (1.2)	8.7 (1.3)	**<0.0001**	12.8 (1.0)	16.6 (1.6)	3.7 (1.8)	0.1576	2.8 (2.0)	**0.0268**	0.65
Hb, g/L	114.4 (1.5)	128.3 (2.4)	13.8 (3.0)	**<0.0001**	118.2 (2.0)	120.9 (3.2)	2.74 (3.9)	0.8925	10.1 (3.3)	**0.0255**	0.56

**^§^**: *p* value from mixed model for repeated measures, interaction group × time, adjusted for admission values. FCM: ferric carboxymaltose; SE: standard error; TSAT: transferrin saturation. Bold: highlight significant *p* values.

## Data Availability

The data are not publicly available as hosted by the restricted hospital database of the Azienda Ospedale Università Padova.

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
