# Peer review of "Ferric Carboxymaltose in Patients with Acute Decompensated Heart Failure and Iron Deficiency: A Real-Life Study"

_jpm, 2023, doi:10.3390/jpm13081250_

Round 1

Reviewer 1 Report

The article “ Ferric carboxymaltose in patients with acute decompensated heart failure and iron deficiency: a real- life study” written by Federico Capone, et al is an original article that aims to identify the clinical significance of iron deficiency screening and iv iron treatment in decompensated heart failure.

Some comments:

1.      The diagnosis of the study patients is unclear. Because the study included all types of HF (Preserved, midrange and reduced EF) the criteria for diagnosis in all these categories should be specified, especially for preserved EF.

2.      Is there a difference between the reduced EF group and preserved EF group regarding hospitalizations or mortality in patients with iron treatment?

3.      Maybe you should highlight the significant p to be more easily detected.

Reviewer 2 Report

In the current work authors evaluate the clinical significance of iron deficiency screening and ferric carboxymaltose treatment in acute decompensated heart failure. There are number of issues in the manuscript that must be addressed before it can be accepted for publication.

1. Manuscript requires extensive English improvement.

2. Patient data is useful for such a analysis however, the sample size is insufficient in the current study to make any solid conclusions. Moreover, conclusions need to clearly highlight that these are preliminary observations and more detailed study is required for strong conclusions.

3. Figure 2,3 should be redrawn for clarity of font.

4. Authors should include significance of this work in the abstract and in discussion. 

Quality of English is not at par for publication and must be improved.

Round 2

Reviewer 1 Report

The changes made to the manuscript are acceptable. 

Reviewer 2 Report

Manuscript is significantly improved.